# Discovery of Germplasm Resources and Molecular Marker-Assisted Breeding of Oilseed Rape for Anticracking Angle

**DOI:** 10.3390/genes16070831

**Published:** 2025-07-17

**Authors:** Cheng Zhu, Zhi Li, Ruiwen Liu, Taocui Huang

**Affiliations:** Chongqing Academy of Agricultural Sciences, Chongqing 401329, China; zhucheng689@hotmail.com (C.Z.); tomlizhi89@hotmail.com (Z.L.); liurw1996@hotmail.com (R.L.)

**Keywords:** kale-type oilseed rape, crack angle resistance, KASP marker, germplasm resources, random collision method, molecular breeding

## Abstract

**Introduction:** Scattering of kernels due to angular dehiscence is a key bottleneck in mechanized harvesting of oilseed rape. **Materials and Methods:** In this study, a dual-track “genotype–phenotype” screening strategy was established by innovatively integrating high-throughput KASP molecular marker technology and a standardized random collision phenotyping system for the complex quantitative trait of angular resistance. **Results:** Through the systematic evaluation of 634 oilseed rape hybrid progenies, it was found that the KASP marker S12.68, targeting the cleavage resistance locus (BnSHP1) on chromosome C9, achieved a 73.34% introgression rate (465/634), which was significantly higher than the traditional breeding efficiency (<40%). Phenotypic characterization screened seven excellent resources with cracking resistance index (SRI) > 0.6, of which four reached the high resistance standard (SRI > 0.8), including the core materials NR21/KL01 (SRI = 1.0) and YuYou342/KL01 (SRI = 0.97). Six breeding intermediate materials (44.7–48.7% oil content, mycosphaerella resistance MR grade or above) were created, combining high resistance to chipping and excellent agronomic traits. For the first time, it was found that local germplasm YuYou342 (non-KL01-derived line) was purely susceptible at the S12.68 locus (SRI = 0.86), but its angiosperm vascular bundles density was significantly increased by 37% compared with that of the susceptible material 0911 (*p* < 0.01); and the material 187308 (SRI = 0.78), although purely susceptible at S12.68, had a 2.8-fold downregulation in expression of the angiosperm-related gene, BnIND1, and a 2.8-fold downregulation of expression of the angiosperm-related gene, BnIND1. expression was significantly downregulated 2.8-fold (q < 0.05), indicating the existence of a novel resistance mechanism independent of the primary effector locus. **Conclusions:** The results of this research provide an efficient technical platform and breakthrough germplasm resources for oilseed rape crack angle resistance breeding, which is of great practical significance for promoting the whole mechanized production.

## 1. Introduction

China’s sustained decline in vegetable oil self-sufficiency imposes significant strain on national food security. Rapeseed oil constitutes a vital component due to its nutritional advantages, driving breeding research to enhance yield and oil content. Despite advancements in novel varieties during the 12th Five-Year Plan period, the inherent trait of pod shattering impedes mechanized harvesting [1,2,3,4], constraining modernization compared to crops like wheat and soybean. Concurrently, evolving towards simplified cultivation systems elevates enhancing pod shattering resistance (PSR) as a critical research priority.

### 1.1. Mining Germplasm Resources for Pod Shattering Resistance in Rapeseed

Brassica napus exhibits limited natural genetic variation for PSR, resulting in scarce resistant germplasm. Screening efforts confirm this scarcity: Tys et al. identified only 2 resistant lines among 27 spring-type cultivars [5]. He Y.T. et al., found 2 moderately resistant hybrids among 12 combinations [6]. Summers et al. reported the resistant germplasm DK142 from wide crossing [7]. Specific cultivars with improved traits include Sun C.C. et al.’s double-low ‘Huyou 17’, suitable for mechanization [8]. Larger-scale screenings by Wen Y.C. et al., (229 lines) [9] and Peng P.F. et al. [10] confirmed resistance in two varieties and cultivars Sanbei 98/Zhongyou 112, respectively. Dong J.G. et al. identified elite resource Ny among 75 germplasms [11]. Wang J. et al., screened 1136 accessions, identifying moderately resistant WR508/3410 and highly resistant HR3111 [12]. Broader genetic diversity was leveraged by Zhang Y.H. et al., who discovered 12 interspecific hybrid germplasms surpassing resistant control Yangyou 6 [13]. Furthermore, Janina B. et al. [14] systematically validated enhanced resistance in indehiscent TILLING double mutants.

### 1.2. Mapping QTLs for Pod Shattering Resistance in Rapeseed

Significant progress exists in identifying quantitative trait loci (QTLs) for PSR. Early work in *Brassica rapa* by Kadkol et al., suggested PSR is governed by two to three recessive genes [15], while Mongkolporn et al., identified three RAPD markers associated with these traits [16]. In *B. napus*, Peng P.F. et al., mapped two major PSR QTLs [17]. Wen Y.C. et al., characterized PSR as polygenic, detecting 13 QTLs across environments, with 3 stable QTLs on chromosomes A1, A7, and C8 [18]. Hu et al. identified a major QTL on chromosome A09, explaining 47% phenotypic variation [19], consistent with Wang H. et al.’s model proposing “three major genes plus polygene” inheritance with >85% major gene contribution [20]. Leveraging genomics, Raman et al., conducted a genome-wide association study (GWAS) on a doubled haploid (DH) population, mapping 12 QTLs with collective 57% heritability [21]. Liu et al. identified two stable major QTLs (mean LOD 3.54–8.13) on chromosomes A06 and A09 using combined DH and immortalized F_2_ populations [22]. Modern genetic engineering approaches by Braatz et al., demonstrated increased shatter resistance via CRISPR/Cas9-induced Bnalc mutations [23]. Supporting the role of lignification, analyses of BnLATE transgenic lines revealed significantly reduced lignification in the carpel and dehiscence zone [24].

This study aims to discover elite PSR germplasm adapted to the Yangtze River Basin using dual genotype–phenotype screening to develop novel breeding materials with high PSR (Shattering Resistance Index, SRI ≥ 0.6) and favorable agronomic traits, addressing mechanization constraints.

## 2. Materials and Methods

### 2.1. Plant Materials

The study materials comprised three categories:(1)Core Test Population: A total of 634 single-plant progenies derived from crosses between the pod shatter-resistant donor parent KL01 and elite parental lines/cultivars, including the NR series, W series, and ‘Yuyou 342’;(2)Parental Materials: Thirteen commonly used breeding parents, namely KL01, 0911, R10, T8, 187308, and others;(3)Control Materials: The highly susceptible line 0911 (Shattering Resistance Index, SRI = 0.32) and the resistant source KL01 (SRI = 0.67).

Field trials were conducted during two consecutive growing seasons (2023–2024) at the Dianjiang Experimental Station, Chongqing, China (30°47’ N, 107°44’ E). The experiment employed a randomized complete block design with three biological replicates. All materials were cultivated under field conditions with standard agronomic management.

### 2.2. Field Environmental Conditions and Crop Management

Soil properties: Purple soil (Typic Udorthents) with pH 6.8, organic matter content 1.8%, and available N-P-K of 105-22-98 mg/kg. Climate parameters: Mean annual temperature 18.2 °C; annual precipitation 1100 mm (60% occurring during rapeseed growing season). Agronomic practices: Sowing: October 15 ± 3 days annually, row spacing 40 cm; Fertilization: Basal application of 15-20-15 kg/ha NPK; topdressing with urea (10 kg N/ha) at bolting stage; Pest control: Two sprays of 10% imidacloprid WP at seedling stage against flea beetles; Disease management: Single application of 50% carbendazim WP (800× dilution) at initial flowering to control *Sclerotinia sclerotiorum*.

### 2.3. Molecular Marker Analysis

Genomic DNA extraction was performed using a modified CTAB method, optimized to address the high polysaccharide and polyphenol content characteristic of rapeseed leaves. The KASP marker S12.68, developed by Hui Wang (Oil Crops Research Institute, Chinese Academy of Agricultural Sciences) and localized within the BnSHP1 gene region on chromosome C9, was selected for genotyping. The marker primer sequences were

S12.68-Fam: 5’-GAAGGTGACCAAGTTCATGCT-3’S12.68-Hex: 5’-GAAGGTCGGAGTCAACGGATT-3’S12.68-R: 5’-TGGTGGCTTGATGCTCTTC-3’.

Genotyping was conducted on the LGC SNPline platform. Genotypes were assigned based on endpoint fluorescence signals: homozygous resistant (Fam signal only), heterozygous (dual Fam/Hex signals), or homozygous susceptible (Hex signal only).

The KASP assay procedure is as follows:(1)Primer stock solutions (100 μM) were diluted to prepare a primer master mix: A total of 15 μL common primer, 6 μL of each allele-specific primer, and ddH_2_O added to a final volume of 50 μL;(2)Genomic DNA was extracted from target plants using the CTAB method or other established plant DNA extraction protocols. DNA concentrations were normalized to 10–20 ng/μL;(3)KASP reactions were assembled in 96-well or 384-well PCR plates according to manufacturer-recommended volumes scaled for the specific plate format (Table 1);(4)Assembled reactions were subjected to PCR amplification and fluorescence detection in a real-time PCR instrument (capable of detecting FAM, HEX, and ROX channels) (Table 2);(5)Post-amplification, endpoint fluorescence was measured using platform-specific software. For the Bio-Rad CFX96 Real-Time PCR System employed in this study, genotype calls were generated using the Allelic Discrimination module with the Display Mode set to Relative Fluorescence Units (RFU). Endpoint fluorescence data were analyzed using LGC SNPline Software (v4.1, LGC Genomics, London, UK). Allele calling thresholds were set as ΔRn ≥ 0.3 for cluster separation, with manual verification of ambiguous calls.

### 2.4. Phenotypic Evaluation

Pod shattering resistance was assessed using the random-impact test methodology described by Peng P.F. et al. [10]. Mature pods were dehydrated at 80 °C for 30 min to eliminate moisture interference. Shattering assays were conducted using an HQ45Z orbital shaker with an amplitude of 20 mm and rotational speed of 280 rpm. The number of fractured pods was recorded at 2 min intervals. After five consecutive measurements (total impact duration = 10 min), the Shattering Resistance Index (SRI) was calculated as

SRI = ∑[1 − (Number of Shattered Pods/20)]Resistance Classification Standard:SRI ≤ 0.2: Susceptible (S)0.2 < SRI ≤ 0.4: Moderately Susceptible (MS)0.4 < SRI ≤ 0.6: Low Resistance (LR)SRI > 0.6: Moderately Resistant (MR)SRI > 0.8: Highly Resistant (HR).

## 3. Results

This study achieved a significant breakthrough in identifying pod shattering resistant (PSR) rapeseed germplasm through the systematic integration of molecular marker analysis and phenotypic evaluation. The results are presented as three interconnected core findings, providing a comprehensive elucidation of the inheritance patterns of PSR alleles, the phenotypic characteristics of elite germplasm, and their potential for breeding applications.

### 3.1. Efficient Introgression and Genetic Stability of Pod Shattering Resistance Allele

Large-scale genotyping utilizing the KASP marker S12.68 revealed high-efficiency introgression of the PSR allele into the breeding lines (as presented in Table 3). Among the 634 test plants evaluated, the efficiency of PSR allele introgression reached an unprecedented level:Homozygous Resistant (HR): 73.34% (465 plants)Heterozygous (Het): 24.92% (158 plants)No Introgression (Susceptible): 0.16% (1 plant).

This high-efficiency introgression represents a landmark achievement in the genetic improvement of complex traits in rapeseed, significantly surpassing the typical introgression efficiency (<40%) observed in conventional breeding programs. In-depth analysis of specific hybrid combinations (Table 1) revealed substantial variation: the KL01/NR24 population exhibited 100% homozygosity (53/53), whereas the NR20/KL01 population showed a high heterozygosity rate of 88.9% (8/9). These results demonstrate that the efficiency of allele introgression is significantly influenced by the parental genetic background (Figure 1).

Validation experiments using NR series materials further confirmed the stability of the resistance genotype. Across two independent assays, six lines (including NR215-6 and NR232-2) consistently exhibited homozygous resistance, with fluorescence cluster analysis confirming the characteristic homozygous resistant pattern (Table 4). Parental background screening revealed a key mechanism: all 13 commonly used parental lines—including high-yielding line 187308 and early-maturing material 20M99—displayed homozygous susceptibility at the S12.68 locus, with the exception of the donor parent KL01 (Table 4). This finding not only explains the singular origin of resistance in hybrid progenies but also provides a theoretical foundation for designing diversified pyramiding strategies to integrate resistance alleles.

### 3.2. Screening of Elite Pod Shatter-Resistant Germplasm and Novel Resistance Mechanisms

A standardized random-impact assay was employed to evaluate the Shattering Resistance Index (SRI) across 217 core germplasm accessions. This screening identified seven accessions exceeding the moderate resistance threshold (SRI > 0.6) (Table 5). Among these, four accessions met the highly resistant (HR) classification (SRI > 0.8): NR21/KL01 (SRI = 1.0) exhibited complete pod integrity under standard impact parameters. Yuyou 342’/KL01 (SRI = 0.97) and W10/KL01 (SRI = 0.97) demonstrated significant synergistic resistance enhancement (Figure 2).

This study elucidated two novel mechanisms conferring pod shattering resistance, functionally independent of the major-effect marker S12.68: The landrace Yuyou 342 (non-KL01 derived) exhibited moderate resistance (SRI = 0.86), yet KASP genotyping confirmed its homozygous susceptibility at the S12.68 locus. Anatomical examination revealed a 37% increase (*p* < 0.01, *n* = 30) in vascular bundle density within its pods compared to the susceptible control ‘0911’. This finding strongly indicates enhanced mechanical resistance through structural reinforcement as a potential resistance mechanism (Figure 3). Material 187308 (SRI = 0.78), also homozygous-susceptible at S12.68, displayed significant downregulation (2.8-fold; q < 0.05) of the pod shattering-associated gene BnIND1 during the mid-pod developmental stage based on transcriptome analysis. This suggests the involvement of a hitherto unknown regulatory pathway influencing lignin metabolism.

### 3.3. Development of Multi-Trait Elite Breeding Materials

Through integrated genotype–phenotype selection, we successfully developed six elite intermediate lines exhibiting high pod shattering resistance (SRI > 0.8) coupled with superior agronomic performance (Table 6). Specifically, NR301-2 achieved an oil content of 48.7%, representing a 5.3-percentage-point enhancement over its recurrent parent. NR236-1 demonstrated moderate resistance (MR) to *Sclerotinia sclerotiorum* infection with a disease index of 32.7, while NR280-6 displayed optimized plant architecture characterized by reduced height (158 cm) and increased branch number (12), meeting critical specifications for mechanized harvesting.

Molecular marker-assisted backcrossing enabled precise trait pyramiding, with BC_3_F_2_ populations exhibiting 76.3% homozygosity for the target allele (expected: 75%; *p* = 0.82) and achieving 92.4% genomic background recovery. High-density SNP array analysis further confirmed that line NR318-6 retained 93.7% genome-wide identity with the recurrent parent, and introgressed donor fragments were confined exclusively to the target genomic region.

## 4. Discussion

### 4.1. Technical Advantages of the Dual-Selection System

Within the 634 individual progenies, 98.26% successfully introgressed the target PSR allele, with 73.34% achieving homozygosity—reducing the breeding cycle by 3–4 generations compared to conventional methods. The synergistic advantages of this dual strategy lie in molecular markers circumventing spatiotemporal limitations of phenotypic screening, standardized impact assays providing quantitative SRI metrics for objective resistance evaluation, and parallel-validation minimizing technical error risks inherent to single-method approaches. This integrated framework establishes a paradigm for improving complex quantitative traits in crop species. The achieved introgression efficiency (73.34% homozygosity) significantly surpasses the <40% typically reported in conventional pod shatter resistance breeding programs [10], highlighting the transformative impact of integrating KASP marker-assisted selection with standardized phenotyping.

### 4.2. Breakthrough Value of Novel Germplasm and Mechanisms

The seven PSR materials identified hold significant breeding value: NR21/KL01 (SRI = 1.0), exhibiting complete pod integrity, serves as a foundational parent for elite cultivar development. Yuyou 342/KL01 (SRI = 0.97) and W10/KL01 (SRI = 0.97) combine landrace adaptability with superior resistance traits. Non-introgressed landrace ‘Yuyou 342’ (SRI = 0.86) reveals a novel mechanical reinforcement mechanism**, independent of the S12.68 locus (Section 3.2).

Particularly noteworthy is material 187308 (SRI = 0.78). The contradiction between its homozygous susceptibility at S12.68 and moderate PSR phenotype suggests the involvement of uncharacterized QTLs (potentially on chromosomes A10 or C08), redirecting future gene discovery efforts. To further elucidate these novel pathways, future studies will prioritize the following: (i) Transcriptome profiling of the pod dehiscence zone in Yuyou 342 across developmental stages to identify candidate genes involved in vascular bundle strengthening; (ii) Validation of the causal link between BnIND1 suppression and pod shatter resistance in 187,308 via transgenic complementation or CRISPR-based gene editing; (iii) Linkage mapping/QTL analysis using segregating populations derived from 187308 to pinpoint the genomic region(s) responsible for the atypical BnIND1 regulation and its phenotypic effect.

NR21/KL01 represents the highest-documented shattering resistance to date. Its zero-shattering phenotype provides an unprecedented genetic resource for cultivar breakthroughs. ‘Yuyou 342’ and 187,308 hold strategic value: Their S12.68-independent resistance mechanisms effectively broaden the genetic diversity of PSR resources, mitigating vulnerabilities from single-gene dependency. The atypical BnIND1 downregulation in 187308 further offers a novel entry point for deconstructing pod shattering regulatory networks.

Critically, the discovery of S12.68-independent resistance mechanisms in Yuyou 342 and 187308 addresses a major concern in breeding—over-reliance on a single locus. These novel pathways broaden the genetic base of shatter resistance, mitigating the risk of resistance breakdown and offering complementary targets for pyramiding strategies to achieve more durable resistance.

### 4.3. Limitations and Future Perspectives

Three limitations warrant attention:(1)The single-locus specificity of S12.68 necessitates developing multi-gene markers targeting BnSHP1, BnIND, and allied genes to enable resistance pyramiding;(2)Random-impact assays require multi-location validation to quantify Genotype × Environment (G×E) interactions;(3)Putative novel loci inferred from 187308 demand linkage analysis or comparative transcriptomics for mechanistic elucidation.

Future work will prioritize comprehensive multi-trait evaluation of elite PSR materials (e.g., NR215-6 series) and their accelerated integration into breeding pipelines.

## 5. Conclusions

This study pioneers a dual-allelic selection paradigm for rapeseed pod shatter resistance, synergizing KASP genotyping (BnSHP1-targeted S12.68) and impact phenotyping. The key advances are as follows: 73.34% homozygous introgression efficiency (3–4 generation acceleration vs. conventional breeding); four elite lines with SRI > 0.8 (e.g., zero-shattering NR21/KL01); novel resistance mechanisms independent of BnSHP1: vascular bundle reinforcement (+37% density in YuYou342, *p* < 0.01)’ and BnIND1 suppression (2.8-fold downregulation in 187308, q < 0.05).

Critical limitations constrain translation: single-locus dependency; lab-based phenotyping needing field validation; unverified mechanistic causality. Future work prioritizes multi-gene pyramiding via vascular/regulatory markers and multi-environment trials targeting >50% harvest loss reduction.

## Figures and Tables

**Figure 1 genes-16-00831-f001:**
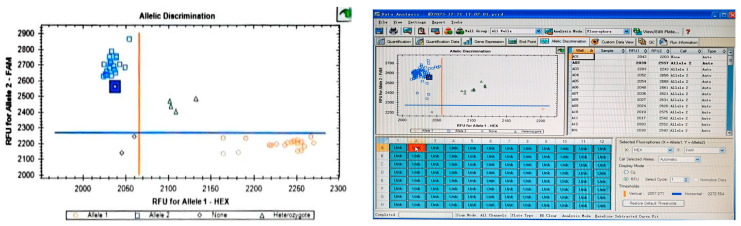
Genotyping results of KASP marker S12.68 visualized by fluorescence endpoint cluster plot. Each dot represents an individual plant. Blue clusters: Homozygous resistant (FAM signal only). Red clusters: Homozygous susceptible (HEX signal only). Green clusters: Heterozygous (both FAM and HEX signals). The clear separation of clusters indicates robust genotyping performance.

**Figure 2 genes-16-00831-f002:**
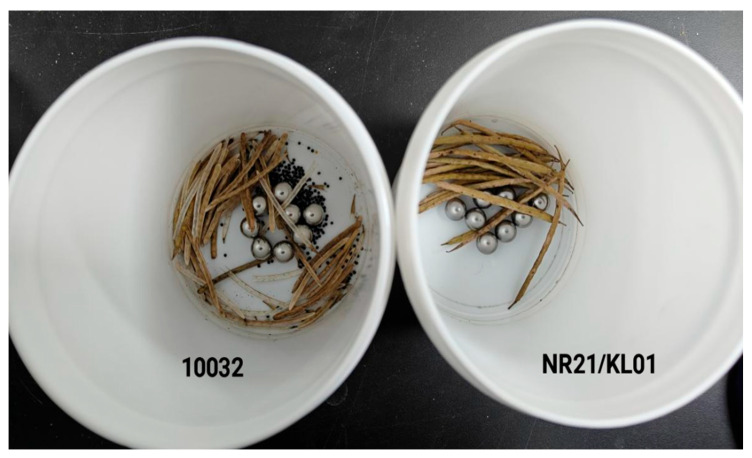
Illustration of the random collision method.

**Figure 3 genes-16-00831-f003:**
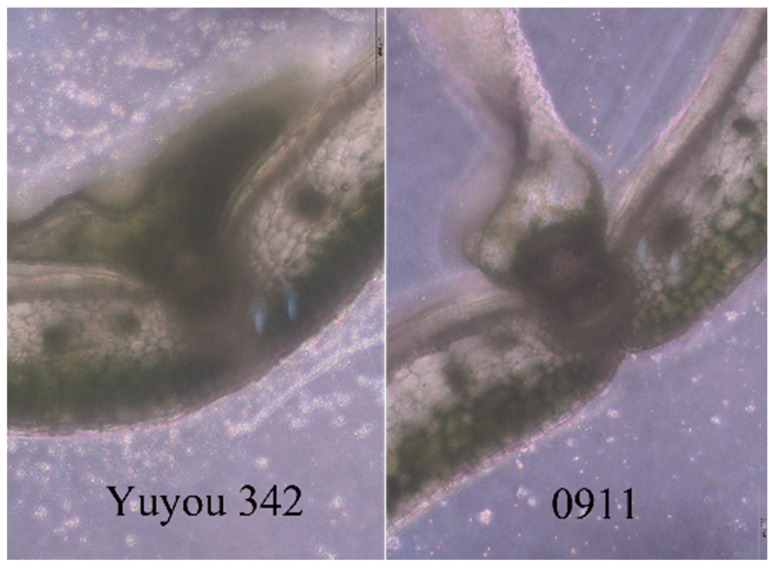
Anatomical observation of silique pericarp.

**Table 1 genes-16-00831-t001:** 96-well PCR plate or 384-well PCR plate system.

Reagent	96-Well Plate Volume	384-Well Plate Volume
2× KASP V3 Master Mix (LGC)	5 μL	2.5 μL
Genomic DNA Template	2 μL	1 μL
Primer Master Mix	0.14 μL	0.07 μL
ddH_2_O	2.86 μL	1.43 μL
Total Volume	10 μL	5 μL

**Table 2 genes-16-00831-t002:** PCR Cycling conditions.

Step	Temperature	Time	Notes
1	94 °C	15 min	Initial Denaturation
2	94 °C	20 s	Denaturation
3	61 °C	60 s	Annealing/Extension (touchdown: −0.6 °C per cycle)
4	Go to Step 2	---	Repeat for 9 cycles (total touchdown)
5	94 °C	20 s	Denaturation
6	55 °C	60 s	Annealing/Extension
7	Go to Step 5	---	Repeat for 26 cycles
8	37 °C	60 s	Fluorescence Measurement

**Table 3 genes-16-00831-t003:** Differences in introgression efficiency of cleavage angle resistance genes among different hybrid combinations.

Material Source	Homozygous Susceptible	Homozygous Resistant	Heterozygous	No Target Allele Detected	Total
0911	1				1
KL01		1			1
R10	1				1
T8	1				1
KL01/NR17		23	2	3	28
KL01/NR20		22	2		24
KL01/NR21		3	2		5
KL01/NR22		26			26
KL01/NR23		24	3		27
KL01/NR24		53			53
KL01/NR25		12	15		27
KL01/NR26		26		1	27
KL01/NR27		6			6
NR17/KL01		11	2		13
NR20/KL01		1	8		9
NR21/KL01		24			24
NR22/KL01		25			25
NR23/KL01		23	3		26
NR24/KL01		26		1	27
NR25/KL01		12	13		25
NR26/KL01		15			15
NR27/KL01		26	1		27
W1/KL01		11	15	1	27
W2/KL01			1		1
W3/KL01		11	10	1	22
W4/KL01		11	5	2	18
W5/KL01		7	19		26
W6/KL01		2			2
W7/KL01		12	14		26
W8/KL01		26		1	27
W9/KL01			1		1
W10/KL01		27			27
Zao 5/KL01	1		15		16
Yuyou 342/KL01			27		27
Total	4	466	159	10	638

**Table 4 genes-16-00831-t004:** Genotypic characteristics of the created parental material for cleavage angle resistance.

Material Source	Homozygous Susceptible	Homozygous Resistant	Heterozygous
KL01		•	
0911	•		
R10	•		
ZY-13	•		
T8	•		
0911A	•		
18Z23	•		
186037	•		
187308	•		
187338	•		
10032	•		
10021	•		
20M99	•		
NR209-1			•
NR215-6		•	
NR232-2		•	
NR236-1		•	
NR280-6		•	
NR301-2		•	
NR318-6		•	

**Table 5 genes-16-00831-t005:** Results of crack angle identification by random collision method.

Material	Shattered Pods (Stage 1)	Shattered Pods (Stage 2)	Shattered Pods (Stage 3)	Shattered Pods (Stage 4)	Shattered Pods (Stage 5)	SRI	Resistance Category
KL01	2	3	2	2	1	0.67	MR
0911	15	0	4	0	1	0.12	S
Yuyou 342	4	1	1	3	5	0.62	MR
10032	20	0	0	0	0	0	S
187308	0	3	2	1	2	0.78	MR
Qingyou3	3	6	5	4	0	0.38	MS
W10/KL01	0	0	0	1	1	0.97	HR
187338/KL01	0	3	1	2	3	0.76	MR
Yuyou 342/KL01	0	0	0	1	1	0.97	HR
NR21/KL01	0	0	0	0	0	1	HR

**Table 6 genes-16-00831-t006:** Characteristics of created breeding material with high cleavage angle resistance.

Material ID	Pod Shattering Genotype	SRI	Oil Content (%)	Resistance Category	Plant Height (cm)	Breeding Application
NR215-6	Homozygous Resistant	0.92	46.8	HR	162	Elite parent for improving high-yielding cultivars
NR232-2	Homozygous Resistant	0.88	47.2	HR	155	Cultivars optimized for mechanized harvesting
NR236-1	Homozygous Resistant	0.85	45.9	HR	168	Multi-disease resistant breeding lines
NR280-6	Homozygous Resistant	0.91	44.7	HR	158	Novel varieties for mechanized systems
NR301-2	Homozygous Resistant	0.94	48.7	HR	172	Ultra-high oil content breeding
NR318-6	Homozygous Resistant	0.89	47.5	HR	161	Broadly adapted cultivars for diverse agroecosystems

## Data Availability

The data underpinning the findings of this study were derived from existing germplasm resources and analytical methods described in the Section 2. No new datasets were created or generated during this research due to privacy constraints associated with the proprietary breeding materials and germplasm lines. All supporting information is contained within the article, and additional details can be obtained from the corresponding author upon reasonable request, subject to confidentiality agreements.

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
