# Peer review of "Discovery of Germplasm Resources and Molecular Marker-Assisted Breeding of Oilseed Rape for Anticracking Angle"

_genes, 2025, doi:10.3390/genes16070831_

Round 1

Reviewer 1 Report

Comments and Suggestions for Authors

Dear Authors, I have reviewed your manuscript. The topic is highly relevant, as pod shattering remains a major barrier to mechanized harvesting of rapeseed. The combination of KASP marker-assisted selection and standardized impact phenotyping is novel and well executed.

Introduction:
The background is clearly presented, emphasizing the importance of improving pod shattering resistance. However, most references are somewhat dated or heavily based on Chinese studies. Including more recent international literature could strengthen the global perspective.

Results:
The results are impressive, particularly the high introgression rates and identification of new resistant lines. The tables are clear, but Figure 1 could benefit from clearer labeling. The discovery of mechanisms independent of S12.68 is especially intriguing and would deserve deeper molecular exploration.

Discussion:
The discussion thoroughly addresses the implications of the findings, though at times it repeats content from the results section. It would help to more clearly separate interpretation from pure data presentation. The section on novel mechanisms is particularly strong and opens exciting future research directions.

Conclusions:
The conclusions effectively summarize the work and highlight its breeding relevance. However, it would be helpful to more explicitly acknowledge the study’s limitations (as partly done already) to provide a balanced close.

Minor points:

  • Be consistent with decimal notation (sometimes using "." and other times ",").

  • A careful language check could polish small English phrasing issues, e.g. “introgressed donor fragments exclusively confined…” could be made smoother.

Reviewer 2 Report

Comments and Suggestions for Authors

The manuscript explains the mechanism of rapeseed pod shattering, which, under certain conditions, is a source of significant yield losses. The Authors have developed a two-pronged ‘genotype-phenotype’ screening strategy. I would like to ask the authors for their opinion on the possible use of the obtained research results in rapeseed breeding. What problems need to be explained most urgently? By what percentage can seed losses be reduced?

Comments

Introduction

In my opinion, it is too long, please shorten it.

Materials and Methods

When was the research conducted?

It is unclear whether this was a field experiment. If so, please include information on the soil and climatic conditions during the experiment and the basic characteristics of the agrotechnology used (sowing dates, fertilisation, chemical protection, etc.).

Full details of the producer of the computer software used in the research?

Results

The tables and figures are clear and understandable.

Discussion

It is rather modest and needs to be expanded.

Conclusion

This is more of a summary than a conclusion. Besides, it is too long.

References

Very few publications (21). Some of the publications were published more than 10 years ago, and even in the 20th century. Please remove them and focus on the most recent ones. A significant number of publications are in Chinese (14/21) and are therefore inaccessible to foreign readers. It is necessary to increase the number of publications in English.
